# Prevalence and Factors Associated with Common Mental Disorders and Posttraumatic Stress Disorder Among Healthcare Workers in a Reference Center for Infectious Diseases During the COVID-19 Pandemic: A Survey-Based Cross-Sectional Study

**DOI:** 10.3390/ijerph22020271

**Published:** 2025-02-13

**Authors:** Patrícia Guimarães, Raquel Oliveira, Rocicley Amud, Maria Elisa Bezerra, Paula Rigolon, Eunes Milhomem, José Luiz Lessa, Guilherme Calvet, Sonia Passos

**Affiliations:** 1Oswaldo Cruz Foundation, Evandro Chagas National Institute of Infectious Diseases, Rio de Janeiro 21040-360, Brazil; raquel.vasconcellos@ini.fiocruz.br (R.O.); rocicley.amud@ini.fiocruz.br (R.A.); elisambezerra2@gmail.com (M.E.B.); rigolonpaula@gmail.com (P.R.); eunes.castro@ini.fiocruz.br (E.M.); sonialambert@gmail.com (S.P.); guilherme.calvet@ini.fiocruz.br (G.C.); 2Institute of Psychiatry, Federal University of Rio de Janeiro, Rio de Janeiro 22290-140, Brazil; jlmlessa@gmail.com

**Keywords:** COVID-19, common mental disorders, posttraumatic stress disorder, GHQ-12, IES-R

## Abstract

A cross-sectional survey conducted between July and September 2020 and August and September 2021 estimated the prevalence and predictors of common mental disorders (CMDs) and post-traumatic stress disorder (PTSD) among healthcare workers (HCWs) at an infectious disease referral center during the COVID-19 pandemic. CMDs were assessed using the General Health Questionnaire (GHQ-12), and PTSD using the Impact of Event Scale-Revised (IES-R). Multivariate logistic regressions identified predictors of CMD and PTSD. The study included 371 HCWs in 2020 and 167 in 2021. The CMD prevalence was 34.3% (95% confidence interval (CI): 29.5–39.4) in 2020 and 30.5% (95% CI: 23.7–38.1) in 2021. The PTSD prevalence was 25.4% (95% CI: 21.0–30.2) in 2020 and 32.7% (95% CI: 25.6–40.5) in 2021. Factors increasing CMD odds included lower education (adjusted odds ratio (AOR) = 3.71, 95% CI: 1.60–8.61), lack of physical activity (AOR = 2.23, 95% CI: 1.33–3.73), and experiencing COVID-19 symptoms (AOR = 1.64, 95% CI: 1.02–2.64) in 2020; chronic diseases (AOR = 3.14, 95% CI: 1.34–7.35) and SARS-CoV-2 testing (AOR = 3.39, 95% CI: 1.13–10.17) raised CMD odds in 2021. Frontline HCWs had lower CMD odds in 2020 (AOR = 0.60, 95% CI: 0.36–1.00) and 2021 (AOR = 0.33, 95% CI: 0.14–0.75). PTSD was associated with having COVID-19 symptoms (AOR = 2.06, 95% CI: 1.23–3.45), living with high-risk individuals (AOR = 1.75, 95% CI: 1.03–2.95), and losing loved ones (AOR = 1.86, 95% CI: 1.09–3.17) in 2020. Chronic diseases increased PTSD odds in 2020 (AOR = 2.20, 95% CI: 1.25–3.86) and 2021 (AOR = 2.30, 95% CI: 1.03–5.13), while age decreased them in 2020 (AOR = 0.95, 95% CI: 0.93–0.98) and 2021 (AOR = 0.95, 95% CI: 0.91–0.99). Younger HCWs with chronic conditions living with high-risk individuals were particularly affected. These findings highlight the urgent need for targeted emotional support and stress management programs for HCWs.

## 1. Introduction

Previous studies have shown increased mental disorders, particularly symptoms of depression and anxiety, among healthcare workers (HCWs) who have experienced severe acute respiratory syndrome (SARS) and Middle East respiratory syndrome (MERS) [1,2,3,4,5]. Additionally, posttraumatic stress disorder (PTSD) was observed in the population of HCWs during these two epidemics [6,7,8].

Anxiety and depressive disorders are referred to as common mental disorders (CMDs) and are highly prevalent in the general population [9]. In Brazil, studies indicate that the prevalence of nonpsychotic CMDs in the general population ranges from 7% to 26%, with an average of 17% [10].

PTSD is a set of symptoms associated with a highly traumatic stressor to one’s physical integrity or by a family member or other person in close association with the individual, involving death, severe injury, or other threats. The response to the event includes intrusive memories, avoidance, hyperarousal, and mood or cognitive alterations [11].

At the beginning of the pandemic (2020), health professionals mobilized to provide treatment and support resources to the population in a climate of uncertainty and emotional and physical stress. Hospitals quickly became crowded, and several reports of personal protective equipment (PPE) shortages emerged. In addition to the lack of specific treatment for the disease, fear for one’s own life and that of loved ones, fear of being socially excluded for being associated with the disease, and finally, dealing with the suffering and death of similar people in serious proportions were, for a long time, questions that were present in the daily lives of HCWs [12].

All this contributed to an increase in the psychic suffering of these professionals. Reports of fatigue, forgetfulness, irritability, difficulty concentrating, and somatic complaints such as insomnia, feelings of worthlessness, and low self-esteem were common. Such symptoms could significantly impact the mental health of these professionals [13].

After the availability of vaccines against COVID-19, the appearance of new variants of the SARS-CoV-2 virus [14], increasing reports of reinfections [15], and recognition of the development or persistence of signs and symptoms after acute COVID-19 (long COVID-19) [16], a new scenario emerged with novel challenges and potential impacts on the mental health of HCWs since early 2021.

An umbrella review of forty-four published meta-analyses of observational studies evaluating the prevalence of mental health problems among hospital staff during the pandemic reported that one-third of hospital workers reported anxiety and depression symptoms. Additionally, female hospital workers reported more anxiety and depression than their male counterparts, and doctors’ posttraumatic stress symptoms were greater than those of nurses [17].

### Aims

The main objective of this study was to estimate the prevalence of and factors associated with common mental disorders and probable posttraumatic stress disorder among HCWs, providing an important contribution to existing literature with large, validated instruments and a population that was stratified to compare two representative groups (frontline HCWs and non-frontline HCWs) in two different phases of the pandemic (2020 and 2021). This study is particularly unique because the HCWs were enrolled in a national institute dedicated solely to infectious diseases, providing a distinct perspective on the psychological effect of the pandemic within a specialized healthcare setting.

## 2. Materials and Methods

A serial cross-sectional study was conducted during two periods of the COVID-19 pandemic in Rio de Janeiro, Brazil: from July to September 2020 (referred to as period one survey in this study) and from August to September 2021 (referred to as period two). The study involved the self-completion of structured questionnaires and validated screening instruments for mental disorders.

The sample calculation included a minimum of 385 professionals required to identify a 50% prevalence of mental disorders, with a 10% absolute error at the 95% confidence level. Enrollment was consecutive until the minimum sample size was reached among the approximately 1800 health professionals at the Evandro Chagas National Institute of Infectious Diseases (INI), including the hospital center for hospitalized patients with COVID-19. The sample size was calculated using WinPepi software version 9.7 [18].

The study population included adults aged 18 years and older who were HCWs at the INI, Oswaldo Cruz Foundation, Rio de Janeiro, Brazil.

### 2.1. Data Collection and Validated Instruments

Structured questionnaires (Appendix A) were emailed to participants using REDCap version 14.5.31 (Research Electronic Data Capture tool) [19,20]. Up to three reminder messages were sent to potential participants to enhance survey completion rates. Researchers contacted various work sectors to mitigate selection bias and ensure the inclusion of workers without regular email access. Additionally, the research project was publicized on the INI webpage.

The structured questionnaires covered various domains, including sociodemographics, comorbidities, professional technical information, occupational data (such as area, sector of daily activities, working hours, and direct contact with COVID-19 patients), lifestyle, and COVID-19-related questions. These COVID-19-related questions included inquiries about self-hospital admission, symptoms, testing, diagnosis, whether professionals were living with a person at high risk of becoming seriously ill from COVID-19, whether they had family members with COVID-19, and whether they had lost a family member or friend to COVID-19.

Validated instruments for screening for CMD and PTSD included the 12-item General Health Questionnaire (GHQ-12) and the Impact of Event Scale-Revised (IES-R), respectively.

The GHQ is widely used in epidemiological surveys to estimate the prevalence of CMD, including anxiety and depression, in various settings, such as primary care outpatient clinics, the general population, and specific groups, such as HCWs [19,20,21].

The 12-item version (GHQ-12) assesses the severity of psychological complaints experienced in the past two weeks. We utilized the binary scoring method (0-0-1-1). Scores were summed to produce a total score ranging from 0 to 12, with higher scores indicating more significant psychological distress [22].

A validated Brazilian version of the instrument [23] was used in the study, employing a cutoff point of 5 to identify more severe and intense symptoms. This decision was influenced by the study sample’s high level of education and the WHO’s recommendation of a 4/5 cutoff point [24].

The IES-R is a PTSD symptomatology screening tool that has good discriminant validity and diagnostic utility [25]. It can be used at any stage of symptom development, whether acute, chronic, or late.

The IES-R is a Likert-type scale designed for self-application in which individuals respond to questions based on their experiences in the seven days preceding the scale administration [26,27]. It comprises 22 items, aligning with the PTSD evaluation criteria outlined in the DSM-IV [11]. The score for each question ranges from 0 to 4 points, with unanswered questions disregarded. The total score is obtained by summing the scores [26,27]. We utilized the validated Brazilian version of the IES-R, which has demonstrated good reliability and validity and has proven effective in screening for posttraumatic stress disorder [28].

One year after the initial survey, the same electronic questionnaires were sent to all HCWs at the institute, including individuals who participated in the 2020 survey and those who did not.

### 2.2. Definitions

A frontline healthcare worker was defined as a professional directly involved in clinical activities, including diagnosis, treatment, or care delivery to outpatients and inpatients with suspected or confirmed COVID-19. This classification was determined based on the professional’s technical background, area of specialization, and daily working sector within our reference institute. Given our institute’s role as a national referral center for infectious diseases, simply accounting for professional titles would not accurately reflect the scope of daily activities and patient care related to suspected or confirmed COVID-19 cases.

### 2.3. Statistical Analysis

All questionnaires and study instruments were exported from the database created by REDCap [29,30] to the IBM SPSS Statistical analysis package, version 22 [31]. Total scores for the GHQ-12 were calculated using a cutoff of 5. Our study population was subsequently analyzed based on two groups: those with a GHQ-12 score ≥ 5 or those with a GHQ-12 score < 5. For the IES-R instrument, a threshold of ≥33 was used to define probable PTSD, and participants were analyzed based on two groups: ’with’ or ’without’ event-related distress [27].

Descriptive statistics were calculated for continuous (presented as medians and interquartile ranges, IQRs) and categorical data. The prevalence of scores on the GHQ-12 and IES-R were calculated along with their respective 95% confidence intervals (CIs). Pearson chi-square or Fisher’s exact tests were employed to assess the associations between categorical variables of interest and scores on the GHQ-12 and IES-R.

Single and multivariate logistic regression analyses were conducted to identify factors associated with two outcomes: the presence of CMD (GHQ-12 score ≥ 5) and probable PTSD (IES-R score ≥ 33). Odds ratios (ORs) and their respective 95% confidence intervals (CIs) were estimated.

The selection of variables for multivariate logistic regression was based on clinical importance and related to COVID-19 from the literature (including sex, age, marital status, having children, self-reported history of chronic diseases, living with a person at high risk for COVID-19, having lost a family member or friend to COVID-19, and being a frontline HCW [32]), and statistical significance (variables significant in the single covariate analysis with *p* < 0.10).

The final multivariate model was constructed using the likelihood ratio test at a significance level of 5%, incorporating the eight selected variables for model adjustment. Additionally, the goodness of fit was assessed using the Hosmer–Lemeshow test [33].

All analyses were stratified by study period (2020: period 1 and 2021: period 2).

## 3. Results

From July to September 2020 and August to September 2021, 371 and 167 professionals, respectively, agreed to participate in the study. In 2021, 82 out of the 167 participants (49.1%) had already participated in the first survey in 2020.

### 3.1. Sociodemographic Characteristics of the Study Population

Excluding marital status and monthly household income, there were no other differences in sociodemographic characteristics between the two survey periods (2020 and 2021), as presented in Table 1. Female subjects predominated in the sample during both study periods (first period, n = 257, 69.3%; second period, n = 123, 73.7%).

During the first and second periods, 192 (51.8%) and 99 (59.3%) professionals, respectively, were considered frontline HCWs, categorized as follows: nurses or nursing technicians (2020 period = 59.9%; 2021 period = 58.6%), physiotherapists (20.8%; 16.2%), physicians (16.7%; 19.2%), nutritionists (1.6%; 2.0%), administrative assistants (1.0%; 1.0%), psychologists (0%; 2.0%), and social workers (0%; 1.0%).

Of the 66 physicians in the study’s first period, 32 (48.5%) were considered frontline HCWs. On the other hand, 87.8% of the 131 nurses/nursing technicians were deemed frontline HCWs.

In the study’s first period, 48.5% of the 60 physicians were considered HCWs. Conversely, 87.8% of the 131 nurses/nursing technicians were deemed HCWs.

### 3.2. Comorbidities, Lifestyles, and Aspects Related to COVID-19 in the Study Population

When completing the questionnaire, 187 (50.4%) and 95 (57.2%) participants reported having at least one clinical comorbidity in the first and second periods of the study, respectively. Eighty-four (22.6%) and seventy (41.9%) health professionals were formally diagnosed with COVID-19, respectively. Additionally, 109 (29.4%) and 19 (11.4%) participants had undergone a SARS-CoV-2 diagnostic test 14 days before completing the survey. One hundred twenty-nine (34.8%) and one hundred sixteen (69.5%) participants had a family member diagnosed with COVID-19. Furthermore, 136 (36.7%) and 98 (58.7%) participants lost a family member or friend, respectively, to the disease in these two periods. Other characteristics were similar between the two study periods and are shown in Table 2.

### 3.3. GHQ-12 and IES-R Scores

The median GHQ-12 score was 3 (IQR: 0–6) in the first period and 2 (IQR: 0–5) in the second period. The prevalence of CMD (GHQ-12 ≥ 5) was 34.3% (95% CI: 29.5–39.4) in period one and 30.5% (95% CI: 23.7–38.1) in period two.

The median IES-R raw score was 19 (IQR: 9–33) in the first period survey and 20 (IQR: 9–36) in the second period survey. The prevalence of probable PTSD (IES-R ≥ 33) was 25.4% (95% CI: 21.0–30.2) in period one and 32.7% (95% CI: 25.6–40.5) in period two.

### 3.4. Factors Associated with the GHQ-12 Score

Detailed single covariate logistic regression analysis tables are provided in the supplemental files (Appendix A). Bivariate analysis revealed significant associations between a GHQ-12 score ≥ 5 during the 2020 survey and female sex, a high school education, a self-reported history of chronic diseases, a lack of regular physical activity, experiencing COVID-19-related symptoms in the 14 days before completing the survey, living with a high-risk person who was seriously ill from COVID-19, and being a frontline HCW (Appendix A). In contrast, during the 2021 survey, bivariate analysis revealed significant associations between GHQ-12 scores ≥ 5 and marital status, a self-reported history of chronic disease, experiencing COVID-19-related symptoms or undergoing a SARS-CoV-2 diagnostic test in the 14 days before completing the survey, experiencing the loss of a family member or friend due to COVID-19, and being a frontline HCW (Appendix A).

Table 3 presents the results of multivariate logistic regression analyses for CMD (GHQ-12 ≥ 5). After controlling confounders, the results indicate that having up to a high school education, not engaging in regular physical activity, and experiencing COVID-19-related symptoms 14 days before completing the survey were associated with increased odds of CMD during the 2020 survey. Conversely, a self-reported history of chronic diseases and undergoing testing for SARS-CoV-2 14 days before completing the study survey were associated with increased odds of CMD during the second survey in 2021. Notably, being a frontline HCW was associated with lower odds of CMD in both periods.

### 3.5. Factors Associated with the IES-R Score

Bivariate analysis revealed significant associations between an IES-R score ≥ 33 during the 2020 survey and female sex, age (per year increase), living alone, self-reported history of chronic disease, experiencing COVID-19-related symptoms in the 14 days before completing the survey, living with a high-risk person who was seriously ill from COVID-19, having family members with COVID-19, and experiencing the loss of a family member or friend due to COVID-19 (Appendix A). Conversely, during the 2021 survey, bivariate analysis revealed significant associations between IES-R score ≥ 33 and female sex, age (per year increase), having children (≤16 years), and not engaging in regular physical activity (Appendix A). The multivariate logistic regression analysis for the scored IES-R included these significant variables and the same confounders listed to construct the factors associated with the scored GHQ-12 models (Table 4).

Factors associated with high odds of probable PTSD (IES-R ≥ 33) during the first survey (2020), after adjusting for confounders, included the presence of COVID-19-related symptoms in the previous 14 days before completing the survey, living with a person at high risk of becoming seriously ill from COVID-19, and experiencing the loss of a family member or friend due to COVID-19. Conversely, not engaging in regular physical activity was associated with low odds of probable PTSD during the second survey (2021). Additionally, for both periods, a self-reported history of chronic diseases was associated with high odds of probable PTSD. In contrast, age was associated with low odds of probable PTSD (less 5% per year increase) (Table 4).

## 4. Discussion

In both periods of our study, we observed a high prevalence of CMD (34.3% and 30.5%) and probable PTSD (25.4% and 32.7%). A meta-analysis [34] including 401 studies involving 458,754 HCWs across 58 countries reported pooled prevalence rates for depression of 28.5%, anxiety of 28.7%, and PTSD of 25.5% [34]. However, it is essential to note that the studies included in the meta-analysis exhibited high clinical and statistical heterogeneity due to variations in settings, countries, populations, and the use of different instruments. The highest prevalence observed in our study was similar to these pooled estimates, and we employed two validated screening tools.

Studies have indicated that young female HCWs are more prone to developing mental disorders [34,35]. Indeed, in the first survey, being female was associated with increased odds of having a GHQ-12 score ≥ 5 in the univariate analysis. Additionally, each additional year of age was found to be a protective factor against probable PTSD in both study periods.

We observed that individuals with up to a high school education were more likely to have CMD during the study’s first period. Lower educational attainment has been linked to poor mental health outcomes during previous outbreaks, such as SARS [2], where lower education resulted in poor mental health among health workers who had taken care of suspected SARS patients [2]. However, studies examining the association between educational levels and mental health outcomes during the COVID-19 pandemic have yielded mixed findings, with some studies finding no significant association and others reporting a potential risk with higher levels of education [36].

During the 2020 survey, COVID-19-related symptoms and lack of regular physical activity were associated with increased odds of developing CMD. Physical activity/exercise was identified as the most common coping behavior among New York HCWs during the COVID-19 pandemic [37]. Another study suggested that moderate physical activity during COVID-19 helped mitigate the adverse effects of the pandemic on psychological health [38]. Individuals who exercise regularly tend to have better mental health and emotional well-being and experience lower rates of mental illness [39].

Factors associated with probable PTSD differed between the two study periods. In 2020, in addition to comorbidities and nonspecific respiratory symptoms, HCWs’ fear for family members at elevated risk of severe COVID-19 or who had lost a family member or friend to COVID-19 emerged as a new factor. Research indicates that HCWs who fear contracting COVID-19, whether for themselves or their loved ones, are more likely to experience PTSD [36]. Siamisang et al. demonstrated that the presence of household members with chronic heart or lung disease, as well as experiencing the loss of relatives or friends to COVID-19, were predictors of stress and anxiety [40].

Interestingly, sedentary behavior emerged as a protective factor against PTSD during the 2021 survey. This finding could be attributed to fears of contracting SARS-CoV-2 in closed environments such as gyms, which may have reduced physical activity levels. Notably, in 2021, new SARS-CoV-2 variants emerged, and reports of reinfection increased despite the availability and use of vaccines [41].

A self-reported history of chronic disease was associated with the presence of CMD in 2021 and probable PTSD in both periods in our study. HCWs with preexisting medical conditions were more likely to develop CMD and PTSD. This susceptibility is likely due to their increased physical vulnerability, fear of COVID-19 infections, and stress associated with the disease, given that these factors are known to enhance morbidity and mortality from COVID-19 [42,43].

Several studies have shown that being a frontline HCW during the COVID-19 pandemic is a risk factor for mental illness [17]. These workers face numerous challenges and stressors that can negatively impact their mental well-being. However, it is essential to note that being a frontline HCW alone does not automatically predict mental distress for every individual. Various factors can influence an individual’s experience and response to a pandemic, including personal resilience, access to support systems, coping strategies, and overall work environment.

Notably, in our study, being a frontline HCW was associated with lower odds of scoring on the GHQ-12 in both study periods, thus serving as a protective factor against the development of CMD. This finding can be attributed to several factors. Notably, nurses and nursing technicians composed almost 60% of the frontline group in our study. Nurses undergo extensive training, equipping them with the skills and knowledge to manage challenging situations effectively. This training enables them to understand risks better, take appropriate precautions, and provide care effectively, thereby potentially reducing anxiety and distress among frontline HCWs.

Moreover, nurses often possess an intense sense of purpose and dedication, driven by a desire to help others and positively impact patients’ lives. This sense of purpose can provide motivation and resilience during challenging times, potentially aiding in their adaptation to the challenges of the pandemic and protecting their mental well-being. Literature studies illustrate how healthcare providers can cope with patient death, even in emotionally charged situations like perinatal loss, find meaning in their work, and maintain a positive outlook, ultimately fostering resilience and mitigating mental health challenges [44,45]. Labrague et al. also demonstrated that psychological resilience reduces the negative impact of compassion fatigue on frontline nurses’ job satisfaction, turnover intention, and quality of care in their assigned units [46]. Additionally, Lekka et al. indicated that resilience levels could have a protective effect against developing PTSD symptoms [47].

Situations such as lack of access to PPE and inadequate support from employers, including insufficient health team training, have been associated with the development of fear, anxiety, depression, sleep disorders, and stress in numerous studies [35]. However, comprehensive support has been provided to our staff since the onset of the pandemic, and a specialized hospital on our campus to manage COVID-19 cases has been established, coupled with our institute’s status as a national reference center for infectious diseases. This support includes the provision of PPE, specialized training, and psychological support. Therefore, it is unlikely that these issues were associated with the outcomes of our study. However, since the surveys did not directly address these aspects with healthcare professionals, we did not include these variables directly in the statistical models.

The COVID-19 pandemic has imposed unprecedented demands on HCWs. A heightened workload can exacerbate PTSD symptoms by increasing arousal and triggering avoidance behaviors [48]. Fortunately, at our institute, all workers had their weekly workload adjusted according to their professions and activities, with designated rest periods.

Health professionals who lack adequate support from their colleagues, family, or friends may experience PTSD symptoms and CMD [36]. Social support can serve as a buffer against the adverse effects of stress. However, in our study, the absence of a social network was not associated with the outcomes.

### Strengths and Limitations

There are strengths and limitations to this study. The strengths of this study include the following: (1) assessment of mental disorders among HCWs over two different periods and scenarios during the COVID-19 pandemic; (2) HCWs conducted activities at the same tertiary healthcare unit, ensuring consistency in sample population characteristics across the two study periods.; (3) as a national reference institute for infectious diseases, our study included HCWs from various professions and sectors, making it possible to categorize two groups of workers as frontline and non-frontline.;(4) consistent questionnaires and validated screening tools were used in both study periods.

The limitations of this study are as follows: (1) we did not investigate the persistence of COVID-19 symptoms after the acute infection or the exact timing of the development of CMD and PTSD symptoms. It has been reported that depression, anxiety, PTSD, and other neuropsychiatric symptoms such as sleep disturbances, fatigue, and cognitive deficits are commonly reported in studies on long COVID [49]. Therefore, the diagnosis of long COVID could have influenced the positive screening scores of the instruments in our study. However, this was probably not the case during the first period of the study, as it was conducted in the early months of the pandemic in Rio de Janeiro; (2) the GHQ-12 is a self-report questionnaire, and its accuracy may be influenced by social desirability or response bias. However, both the GHQ-12 and IES-R are internationally validated tools extensively used for screening CMDs and probable PTSD, and they identified a high prevalence of mental distress among HCWs during the COVID-19 pandemic. If this scenario was underestimated due to possible response bias, the prevalence of both issues could be even greater; (3) additionally, while the IES-R is a valuable screening tool with high sensitivity, some individuals may experience PTSD symptoms unrelated to the COVID-19 pandemic but arising from other traumatic experiences; (4) finally, our study did not aim to evaluate preexisting psychological or mental health conditions, fear of stigmatization, or coping strategies related to COVID-19, which could be associated with the development of mental health problems.

Therefore, these two screening tools should be supplemented with clinical interviews for scored individuals to ensure a comprehensive diagnostic assessment. Effective psychological interventions should also be offered to address individual needs and prevent long-term adverse outcomes.

## 5. Conclusions

In conclusion, this study highlights the psychological burden experienced by healthcare workers in a specialized institute for infectious diseases, underscoring the need to address mental health proactively in healthcare settings. It is essential to develop broader strategies that extend beyond specific crises. Institutions should prioritize mental health by cultivating resilient work environments, conducting regular mental health screenings, and providing accessible support systems. Establishing psychological safety nets, offering targeted stress management training, and ensuring adequate staffing and protective measures can further alleviate healthcare workers’ mental health challenges. These efforts will strengthen healthcare systems’ preparedness for future emergencies while safeguarding the overall mental well-being of their workforce.

## Figures and Tables

**Table 1 ijerph-22-00271-t001:** Characteristics of the participants enrolled in the study by survey period (2020, n = 371; 2021, n = 167).

Characteristics		2020 Survey n (%)	2021 Survey n (%)
Age, Median (IQR)			
	Men	39 (32–45)	41.5 (35–46)
	Women	38 (31–46)	41(33–50)
Highest educational attainment			
	Up to high school	56 (15.1)	22 (13.2)
	University or higher	315 (84.9)	145 (86.8)
Marital status			
	Single	149 (40.2)	49 (29.3)
	Married/living with a partner	172 (46.4)	92 (55.1)
	Separated, divorced, or widowed	49 (13.2)	26 (15.6)
	Missing	1 (0.3)	-
Household monthly income			
	Up to 2 min wage	32 (8.6)	9 (5.4)
	3–7 min wage	166 (44.7)	63 (37.7)
	>8 min wage	137 (36.9)	83 (49.7)
	Did not inform	36 (9.7)	12 (7.2)
Religion			
	No	65 (17.5)	23 (13.8)
	Yes	306 (82.5)	144 (86.2)
Living alone			
	No	306 (82.5)	144 (86.2)
	Yes	65 (17.5)	23 (13.8)
Having children (≤16 years)			
	No	224 (60.4)	100 (59.9)
	Yes	147 (39.6)	67 (40.1)
Social network			
	No	21 (5.7)	9 (5.4)
	Yes	350 (94.3)	158 (94.6)
Profession			
	Nursing technician	73 (19.7)	40 (24.0)
	Physician	66 (17.8)	37 (22.2)
	Nurse	58 (15.6)	28 (16.8)
	Physiotherapist	41 (11.1)	16 (9.6)
	Administrative assistant	20 (5.4)	1 (0.6)
	Biologist	16 (4.3)	8 (4.8)
	Psychologist	14 (3.8)	3 (1.8)
	Pharmacist	12 (3.2)	5 (3.0)
	Laboratory technician	10 (2.7)	4 (2.4)
	Others ^†^	61 (16.4)	25 (15.0)
Weekly working hours			
	<40 h	185 (49.9)	87 (52.1)
	≥40 h	186 (50.1)	80 (47.9)
Development of work activities in the 14 days before completing the survey			
	Exclusively remote	38 (10.2)	11 (6.6)
	Entirely in person	260 (70.1)	123 (73.7)
	Hybrid	73 (19.7)	33 (19.8)
Frontline healthcare workers			
	No	179 (48.2)	68 (40.7)
	Yes	192 (51.8)	99 (59.3)

2020 Survey: Missing age (n = 1); 2021 Survey: Missing age (n = 1). IQR, interquartile range. ^†^ Others: In 2020: one professional each: lawyer, information technology analyst, archivist, biomedical professional, curator, network manager, endemic disease agent, microbiologist, receptionist, pharmacy technician, radiology technician, occupational safety technician, immunohematology technician, metrology technician, health surveillance technician, and zootechnician. Two professionals each: technical support, communication advisory, research assistant, telemarketing operator, and pedagogue. Three professionals each: statistician, nutritionist, and pharmacy technician. Four social workers, seven graduate students, seven veterinarians, and eight financial and administrative management professionals. In 2021: one professional each: archivist, biomedical equipment technician, database analyst, endemic disease agent, financial and administrative management, metrology technician, network manager, research assistant, social worker, telemarketing operator, and zootechnician. Two professionals each: biomedical professional, graduate student, statistician, veterinarian. Three communication advisors and three nutritionists.

**Table 2 ijerph-22-00271-t002:** Comorbidities, lifestyles, and COVID-19-related characteristics by survey period (2020, n = 371; 2021, n = 167).

Characteristics		2020 Survey n (%)	2021 Survey n (%)
Self-reported history of chronic diseases			
	No	184 (49.6)	71 (42.8)
	Yes	187 (50.4)	95 (57.2)
Obesity/overweight			
	No	242 (65.2)	100 (60.2)
	Yes	129 (34.8)	64 (38.6)
	Unknown	-	2 (1.2)
Hypertension			
	No	318 (85.7)	130 (78.3)
	Yes	51 (13.7)	32 (19.3)
	Unknown	2 (0.5)	4 (2.4)
Rheumatologic/joint diseases			
	No	339 (91.4)	151 (91.0)
	Yes	27 (7.3)	14 (8.4)
	Unknown	5 (1.3)	1 (0.6)
Asthma			
	No	347 (93.5)	152 (91.6)
	Yes	22 (5.9)	13 (7.8)
	Unknown	2 (0.5)	1 (0.6)
Diabetes mellitus			
	No	355 (95.7)	152 (91.6)
	Yes	11 (3.0)	9 (5.4)
	Unknown	5 (1.3)	5 (3.0)
Chronic obstructive pulmonary disease			
	No	356 (96.0)	161 (97.0)
	Yes	11 (3.0)	4 (2.4)
	Unknown	4 (1.1)	1 (0.6)
Other chronic diseases ^†^			
	No	341 (91.9)	144 (86.7)
	Yes	30 (8.1)	22 (13.3)
	Unknown	-	-
Current smoking		22 (6.0)	8 (4.8)
Having hobby			
	No	104 (28.1)	55 (33.1)
	Yes	266 (71.9)	111 (66.9)
Having pet			
	No	160 (43.5)	68 (40.7)
	Yes	208 (56.5)	99 (59.3)
Having regular physical activity			
	No	214 (58.8)	88 (53.0)
	Yes	150 (41.2)	78 (47.0)
Admitted to hospital in the 14 days before completing the survey			
	No	363 (97.8)	165 (98.8)
	Yes	8 (2.2)	2 (1.2)
COVID-19-related symptoms in the 14 days before completing the survey			
	No	198 (53.4)	76 (45.5)
	Yes	173 (46.6)	91 (54.5)
SARS-CoV-2 diagnostic test performed in the 14 days before completing the survey			
	No	262 (70.6)	148 (88.6)
	Yes	109 (29.4)	19 (11.4)
Formally diagnosed with COVID-19 by a physician			
	No	282 (77.0)	97 (58.1)
	Yes	84 (23.0)	70 (41.9)
Living with a high-risk person of getting seriously ill from COVID-19			
	No	232 (63.0)	96 (57.5)
	Yes	136 (37.0)	71 (42.5)
Has or have had any family members with COVID-19			
	No	239 (64.9)	51 (30.5)
	Yes	129 (35.1)	116 (69.5)
Have lost a family member or friend to COVID-19			
	No	233 (63.1)	69 (41.3)
	Yes	136 (36.9)	98 (58.7)

2020 Survey: missing: current smoking (n = 7), having a hobby (n = 1), having a pet (n = 3), having regular physical activity (n = 7), formally diagnosed with COVID-19 by a physician (n = 5), living with a high-risk person who is seriously ill with COVID-19 (n = 3), has or have had any family members with COVID-19 (n = 3), have lost a family member or friend to COVID-19 (n = 2); 2021 Survey: missing: self-reported history of chronic diseases (n = 1), having hobby (n = 1), having regular physical activity (n = 1). ^†^ Other chronic diseases: coronary disease, stroke, other heart diseases, liver disease, neurologic disease, cancer, hematologic disease, kidney disease, and thyroid disease.

**Table 3 ijerph-22-00271-t003:** Multivariate logistic regression analysis of factors associated with the GHQ-12 score among healthcare workers (2020 and 2021 survey periods).

Characteristics		2020 Survey					
		Crude OR	95% CI	*p* Value	Adjusted OR ^†^	95% CI	*p* Value
Highest educational attainment							
	Up to high school	3.66	1.67–8.01	0.001	3.71	1.60–8.61	0.002
	University or higher	1	-		1	-	
Having regular physical activity							
	No	2.3	1.45–3.65	<0.001	2.23	1.33–3.73	0.002
	Yes	1	-		1	-	
COVID-19 related symptoms in the 14 days before completing the survey							
	No	1	-		1	-	
	Yes	1.93	1.25–2.99	0.003	1.64	1.02–2.64	0.042
Frontline healthcare workers							
	No	1	-		1		
	Yes	0.49	0.32–0.76	0.002	0.60	0.36–1.00	0.051
**Characteristics**		**2021 Survey**					
		**Crude OR**	**95% CI**	***p* Value**	**Adjusted OR ^‡^**	**95% CI**	***p* Value**
Self-reported history of chronic diseases							
	No	1	-		1	-	
	Yes	2.48	1.21–5.09	0.013	3.14	1.34–7.35	0.008
SARS-CoV-2 diagnostic test performed in the 14 days before completing the survey							
	No	1	-		1	-	
	Yes	2.9	1.10–7.65	0.031	3.39	1.13–10.17	0.029
Frontline healthcare workers							
	No	1	-		1		
	Yes	0.43	0.22–0.85	0.014	0.33	0.14–0.75	0.008

2020 survey: one participant did not complete the GHQ-12 form, and ten had missing values in some variables. Therefore, the final multivariate model included 360 participants. 2021 survey: two participants had missing values on some variables. Therefore, the final multivariate model included 165 participants. GHQ-12, the 12-item General Health Questionnaire; CI, confidence interval; OR, odds ratio; IQR, interquartile range. ^†^ Multivariate logistic model adjusted for sex, age, marital status, having children self-reported history of chronic diseases, living with a high-risk person of getting seriously ill from COVID-19, and having lost a family member or friend to COVID-19. ^‡^ Multivariate logistic model adjusted for sex, age, marital status, having children, living with a high-risk person of getting seriously ill from COVID-19, and having lost a family member or friend to COVID-19.

**Table 4 ijerph-22-00271-t004:** Multivariate logistic regression analysis of factors associated with the IES-R score among healthcare workers (2020 and 2021 survey periods).

Characteristics		2020 Survey					
		Crude OR	95% CI	*p* Value	Adjusted OR ^†^	95% CI	*p* Value
Age (per year increase)							
	Median (IQR)	0.95	0.93–0.98	0.024	0.95	0.93–0.98	0.002
Self-reported history of chronic diseases							
	No	1	-		1	-	
	Yes	1.86	1.15–3.01	0.011	2.2	1.25–3.86	0.006
COVID-19 related symptoms in the 14 days before completing the survey							
	No	1	-		1	-	
	Yes	2.13	1.32–3.43	0.002	2.06	1.23–3.45	0.006
Living with a high-risk person of getting seriously ill							
from COVID-19							
	No	1	-		1	-	
	Yes	1.82	1.13–2.95	0.014	1.75	1.03–2.95	0.038
Have lost a family member or friend to COVID-19							
	No	1	-		1	-	
	Yes	1.73	1.07–2.79	0.026	1.86	1.09–3.17	0.022
**Characteristics**		**2021 Survey**					
		**Crude OR**	**95% CI**	***p* Value**	**Adjusted OR ^‡^**	**95% CI**	***p* Value**
Age (per year increase)							
	Median (IQR)	0.97	0.94–0.99	0.08	0.95	0.91–0.99	0.014
Self-reported history of chronic diseases							
	No	1	-		1	-	
	Yes	1.47	0.75–2.86	0.258	2.3	1.03–5.13	0.041
Having regular physical activity							
	No	0.39	0.20–0.77	0.006	0.33	0.15–0.70	0.004
	Yes	1	-		1	-	

2020 survey: one participant did not complete the IES_R form, and seven participants had missing values in some variables. Therefore, the final multivariate model included 363 participants. 2021 survey: two participants did not complete the IES_R form, and three had missing values on some variables. Therefore, the final multivariate model included 162 participants. IES-R, Impact of Events Scale-Revised; CI, confidence interval; OR, odds ratio; IQR, interquartile range. ^†^ Multivariate logistic model adjusted for sex, marital status, having children, and frontline health workers. ^‡^ Multivariate logistic model adjusted for sex, marital status, having children, living with a high-risk person of getting seriously ill from COVID-19, having lost a family member or friend to COVID-19, and frontline health workers.

## Data Availability

The data supporting this study’s findings are not openly available due to reasons of sensitivity but are available from the corresponding author upon reasonable request. They are located in controlled access data storage at the Evandro Chagas National Institute of Infectious Diseases.

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
