# Peer review of "Prevalence and Factors Associated with Common Mental Disorders and Posttraumatic Stress Disorder Among Healthcare Workers in a Reference Center for Infectious Diseases During the COVID-19 Pandemic: A Survey-Based Cross-Sectional Study"

_ijerph, 2025, doi:10.3390/ijerph22020271_

Round 1
Reviewer 1 Report
Comments and Suggestions for Authors
The study is well-written and methodologically robust. Below are recommendations to enhance the clarity and impact of the manuscript:
Abstract
- Consider rewriting the abstract in a more structured format, explicitly including sections on the background, objectives, methods, results, and conclusions. This will enhance its readability and provide a clearer summary of the study. If the journal does not specifically request a structured abstract, maintain this order for improved organization and clarity.
Aims
- Include a separate section for the study aims to improve clarity.
- Elaborate on the significance of the research by discussing its potential contributions to the existing literature and its relevance in addressing mental health issues among healthcare workers.
Methods
- A table summarizing the instruments used and the key questions from the survey or interview would be a nice add. This will provide a quick reference for readers and improve the section's comprehensibility.
Discussion
- Lines 350–353: This section highlights a commendable and essential aspect of the article: recognizing the positive aspects of healthcare work, even in challenging circumstances like the COVID-19 pandemic. Such recognition is crucial in articles addressing healthcare workers' mental health, as it emphasizes resilience and the capacity to find meaning and positivity despite adversity.
However, these specific lines lack citations. I suggestd adding the following references, which provide valuable insights into similar contexts, specifically in palliative care. These studies illustrate how healthcare providers can cope with patient death (even in emotionally charged situations like perinatal loss), find meaning in their work, and maintain a positive outlook, ultimately fostering resilience and mitigating mental health challenges:
- Dahò, M. (2021). An exploration of the emotive experiences and the representations of female care providers working in a perinatal hospice: A pilot qualitative study. Clinical Neuropsychiatry.
- Nuzum, D., Meaney, S., & O'Donoghue, K. (2014). The impact of stillbirth on consultant obstetrician gynecologists: A qualitative study. BJOG: An International Journal of Obstetrics & Gynaecology, 121(8), 1020–1028.
Incorporating these references will enrich the discussion by highlighting the possibility of fostering resilience and positivity in high-stress healthcare environments, offering a broader perspective on mental health protection strategies.
- Strengths and Limitations: Create a separate section to outline these points clearly.
- Lines 409–410: Move this sentence to the aims section for better alignment and logical flow.
Conclusions
- Lines 412–414: The conclusions appear misaligned with the scope of the investigation, which focused on mental health conditions during the pandemic. While the need for interventions is evident, the pandemic has ended, and the conclusions should address broader strategies for protecting healthcare workers' mental health, irrespective of specific crises.
- Suggestions for future research directions or policy improvements should be moved to the discussion section. In the conclusion section, provide a concise statement summarizing the study’s key findings.
Reviewer 2 Report
Comments and Suggestions for Authors
Please find attached file
